# Setting a Baseline Residential Water Demand Management Solution in Urban Towns of Ethiopia

**Mosisa Teferi Timotewos** [1,*], **Matthias Barjenbruch** [1] **and Beshah M. Behailu** [2]

[1] Department of Urban Water Management, Technical University of Berlin, Gustav-Meyer-Allee 25, 13355 Berlin, Germany

[2] Department of Water Supply and Environmental Engineering, Arba Minch University, Arba Minch P.O. Box 21, Ethiopia

\* Correspondence: timotewos@campus.tu-berlin.de or mosisa78@gmail.com; Tel.: +49-1766-3895-462

**Abstract:** Due to the country's expanding population, inadequate urban water management practices, limited community knowledge of water management, and urbanization, urban water management in Ethiopia is of great importance to the administration of the country. This study draws on a qualitative and quantitative research approach to evaluate the available water supply resources and management techniques in three Ethiopian metropolitan districts, factor in the sustainability of the urban water supply services, and then recommend a workable plan for a sustainable urban water management system. Open-ended and semi-structured questionnaires were used to interview urban water utility officials to reveal important information on water demand management and current water supply services. Documented secondary data analysis and field observations are also used to identify the existing problem in order to draw future suggestions. The findings of this study indicate that some of the common issues with urban water supply systems in towns include outdated water supply infrastructures, rapid population growth and corresponding water demand, high water losses in the distribution system, poor water management practices, and a lack of appropriate institutional framework. The assessments of the three study regions demonstrate that while the percentage of water supply coverage hasn't changed significantly over the previous 10 years, both the number of customers and the overall population growth have increased by nearly 50%. In order to address this, the Ethiopian government would need to put up a lot of effort into developing water use policies and raising consumer understanding of water demand management techniques.

**Keywords:** water demand; water supply; Arba Minch; Ziway; Debre Birhan





## 1. Introduction

Climate variability and the degradation of natural resources are issues that civilization must deal with, either directly or indirectly. The majority of communities around the world have their own traditional methods of handling resource use, even though they may not be scientifically aware of the causes and solutions for alterations in this way. Drinking water resources are part of a natural ecosystem whose quality and quantity are being negatively impacted by climate changes [1] and other anthropogenic activities [2,3]. Besides, the scarcity of these resources is an increasing problem across all continents, with poorer communities most badly affected. Water is intrinsically interlinked with people's daily activities and essential to the resilience of human well-being and survival. However, if these resources are not used wisely, they will become increasingly scarce. Such that many cities in developing countries are unable to provide adequate clean water to their fast-growing populations. Therefore, every water management measure would help us and extend more time of survival.

In Africa, where nearly 60% of the continent's land experiences an arid and semiarid climate, the scarcity of water resources for drinking is becoming more severe [4]. According

to reports, Africa is cited as a water-stressed continent, as 50% of the people who live there must drink water from unsafe water sources [5]. Given its rapid population growth, climate change, and irregular rainfall patterns, Sub-Saharan Africa is one of the regions that are most at risk [6]. The UN-Water Report [7] shows that the number of people in Sub-Sahara without access to clean water has increased by 40% since 2000. Furthermore, this region is also suffering from water stress due to fast-growing urban areas [8]. Despite having relatively rich water resources, Ethiopia, which is located in the Horn of Africa, is experiencing unprecedented population growth, rapid urbanization, and climate change upending the balance between water demand and supply. A study conducted on urban water management in Addis Ababa, the capital city of Ethiopia, confirmed that the increase in population and poor urban water management practices had exerted pressure on the available water resources and water supply for the city [9]. The study demonstrated that the current situation of water supply and demand is not sustainable because of the system's management practices and that an integrated water management practice should be implemented to address unmet water demands in the city. The scarcity of drinking water, particularly in developing countries like Ethiopia, is a crucial problem that requires an immediate solution. Despite the immediacy of the issue, and the potential to apply solutions across the country, there is a shortage of literature and studies done on water management in other urban towns of Ethiopia. However, there are recent studies done on methods for managing water demand in other developing areas. For instance. a survey was conducted in Abuja. Nigeria examines household strategies for coping with adequate domestic water supplies and identifies factors that influence technology choices. As a result, 90% of households use storage techniques, and 82% buy bottled water, while little attention is given to domestic water purification and recycling [10]. This measure also affects water quality if the water is stored for a long period of time. Thus, waterborne diseases caused by bacteria found in contaminated domestic water storage tanks increase the risk of spreading waterborne diseases and can lead to outbreaks of many infectious diseases [11]. Therefore, it is necessary to pay attention to water-saving measures at home and carry out careful home training. Several research studies have also shown that variations in physicochemical water quality parameters due to seasonal changes are making urban drinking water increasingly difficult to manage [12–15] and propose a framework using a model to evaluate current and future water demand and supply gaps in Pakistan. As a result, it reveals that demand-side water demand conservation reduces water usage effectively by 23% and also helps to save money and preserve the environment. Another study in the same country also shows that pricing policies have limited scope to drive households' water consumption patterns, such that suggesting non-pricing strategies such as water-saving campaigns to change people's behaviors toward water conservation [16].

Most of the studies on water demand management (WDM) have been conducted in developed economies and mostly focused on measuring price and income elasticities [17]. However, the relevance of the income elasticity is not applicable to developing countries where there is not enough water for drinking. Such countries, like Ethiopia, care more about having adequate water and managing the drinking water resources they already have by reducing non-revenue water and using other demand-side strategies. The majority of the current water demand management strategies and models of water supply surveillance for urban areas are from developed countries and have significant shortcomings when used directly elsewhere. Though, the differences in socio-economic conditions and the nature of water supply services limit the transferability to the Ethiopian context. According to research from Kenya, water sector reforms have also led to the successful application of water demand management initiatives in some urban areas [18]. Overall, water managers and planners have given high priority to locating, developing, and managing new water resources. As a result, emphasis has been given to the supply side of water development, while demand-side management and improvement in patterns of water use have received less attention [19]. Until recently, little attention has been paid to analyzing the needs themselves (demand-side approach) or ways to mitigate them, which in itself requires a major

paradigm shift [20]. Most of the above-mentioned literature has attended to the determinants of water demand management from both the supply and demand sides [9,10,15–18]. Other studies have focused on assessing price and income elasticities [16] or proposed municipal water supply sector alteration to improve water demand management [15]. However, there remains a dearth of studies wherein water demand management activities were assessed from the residents' side up to the government's side. This study addresses such absences in an innovative way. The locations selected for study vary in their climate conditions and their respective residents' water demand management practices. To assess the scope of demand on all levels, trends over the last 10 years in population growth, urbanization, water supply coverage, and the number of people using the town water supply services in the study areas were also evaluated. The study locations, which have not been analyzed in previous research, provided new insights into demand management techniques across social and climatic variations. This analysis generated suggestions for demand management activities that would ensure the sustainability of water resources, and the proposed solutions were examined alongside existing activities and compared with other countries' management techniques. Triangulating new data from an understudied area with the larger conversations in the field shows the potential of the study to initiate novel approaches to water demand management.

According to the Ethiopian government's policy, the management of water resources shall include all stakeholders, including the private sector, and ensure social equity, system reliability, and sustainability [21]. The policy mainly ensures that ownership is developed throughout the lower tiers and that management autonomy is at the lowest administrative level. However, the communities have very limited awareness of water resource management and determinants of water resources, and less attention was given to alerting the lowest-level stakeholders about demand management. There is a lack of strong monitoring of the status of water supply services and water demand management at the lowest-level water sectors. Given this policy and the larger concerns of access and use, the general objective of the study is to assess the status of the current urban water resource management, water supply services, and demand management activities in three urban towns of Ethiopia and to recommend feasible approaches or practices to manage urban water resources sustainably.

## 2. Study Area

This study was conducted in three urban towns of Ethiopia, each located at different altitudes: Arba Minch (1300 m), Ziway (1600 m), and Debre Birhan (2800 m). Arba Minch is a city found 500 km south of the capital city Addis Ababa, located at geographical coordinates of 6°2′0″ N 37°33′0″ E. Even though the city is categorized as one of the hotter climate areas, it is surrounded by two big natural lakes (Abaya and Chamo) and is considered one of the areas with a good groundwater potential zone in the country. The town has an average annual temperature of 29 °C and average annual rainfall of 900 mm. Ziway is located 163 km southeast of Addis Ababa. The town has a latitude and longitude of 7°56′ N 38°43′ E, an annual rainfall of 700–800 mm, and a mean annual temperature of 20 °C. Debre Birhan is one of the coldest towns in Ethiopia, located 120 km northeast of Addis Ababa. The town has a latitude and longitude of 9°41′ N 39°31′ E, a mean annual temperature of 15 °C, and a mean annual rainfall of 1219.2 mm. The topographical locations of the three towns in Ethiopia are displayed in Figure 1 below.

Ethiopia's precipitation pattern and hydrology are influenced significantly by its topography. Ethiopia is a dome-shaped country with a central highland plateau surrounded by lowlands and separated by deep valleys. The great East African Rift Valley divides the country into the west and east ridges. The highlands receive relatively high rainfall, with run-off flowing in different directions to the surrounding lowlands and, in many cases, crossing international boundaries; hence the country's label as the "Water Tower of East Africa" [22]. But there is no river flow into Ethiopia from neighboring countries.

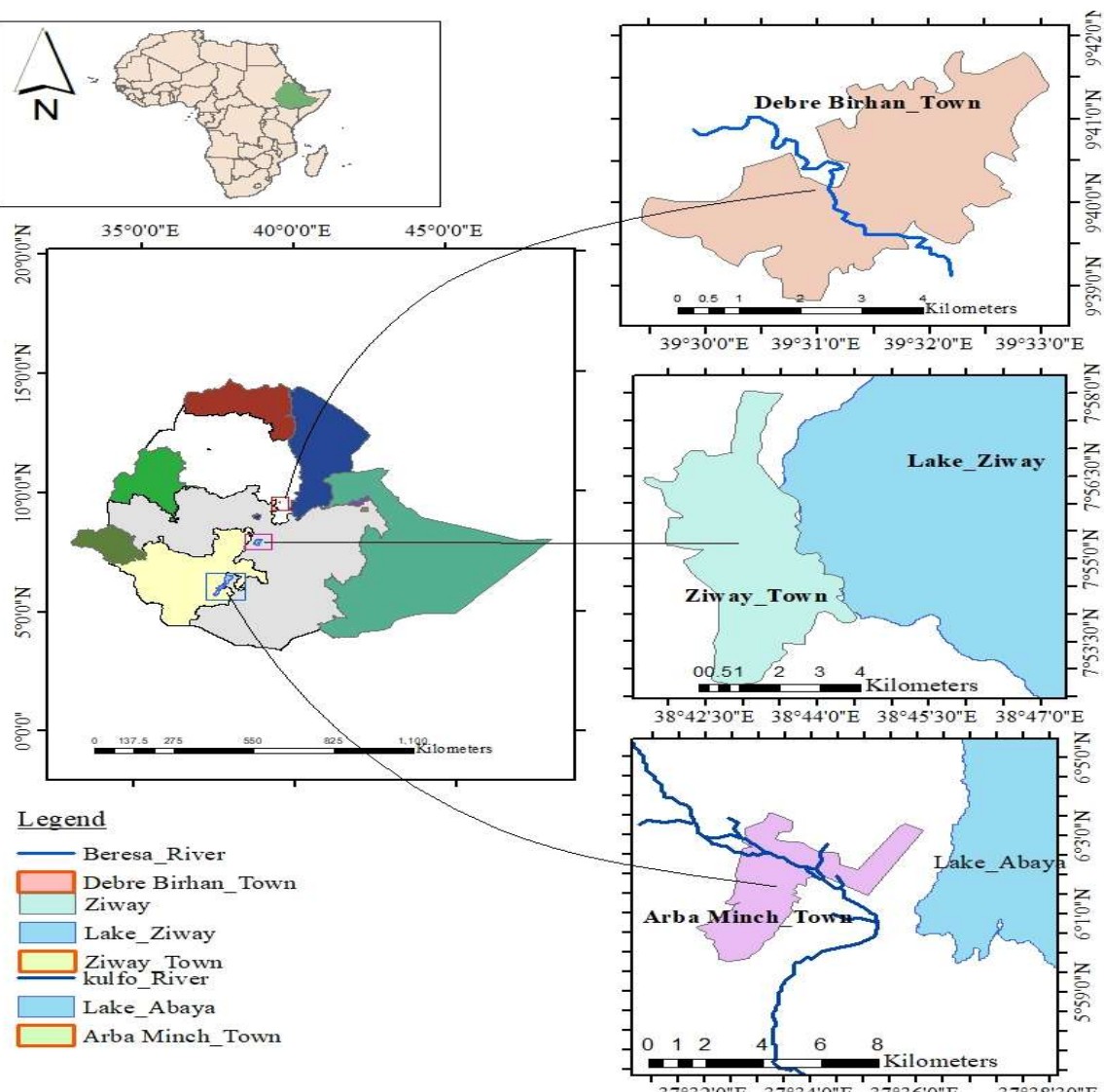

**Figure 1.** Study area location map.

## 3. Data and Methodology

### 3.1. Data

For this study, both qualitative and quantitative data were collected through key informant interviews, field observation, and recorded secondary data analysis. Historical and relevant documented data were collected from the town's Water and Sewerage Authority in the years 2020 and 2021 on current urban water sources, unaccounted-for water or water loss, water distribution, trends of water demand, production, supply coverage, and demographic data. The data was collected from the Water and Sewerage Authority offices in Arba Minch, Debre Birhan, and Ziway Towns, respectively. Studies already done on related topics are also an important source of data used in this study. Any information which analyzed the status of the water supply system and at what stages urban water demand management appeared was reviewed.

### 3.2. Methodology

This study used qualitative and quantitative methods, which employ interpretative and descriptive procedures offering a wide range of understanding of the fact. Qualitative narratives provide flexibility to the subject under study [23], and the quantitative research approach is primarily used to qualify and compare empirical data [24]. Key informant

interviews were conducted in order to collect information about the general status of water supply and its services within their respective towns. Nine water supply experts, 3 from each urban town, were purposely selected based on their professional skills, such that these selected persons are water supply distribution operators, water supply conservation officials, and those responsible for the monitoring and management of urban water supply services. Semi-structured and unstructured open-ended questionnaires were used for the detailed interview conducted. The respondents were interviewed on their personal experiences and current status on general water supply sources and resources, existing regulations on water management, problems and challenges of current water management activities, communities' perceptions and active engagement in water management activities, and their personal experience suggestion needed for the future on enhancing demand management of their respective town. The duration of the interviews lasted 45 min for each expert, and notes memorizing their ideas were taken during the interviews. As the interviews were qualitative and explanatory in nature, data were analyzed following the approach of qualitative data analysis.

Empirical secondary data of the past 10 consecutive years were collected from each urban water utility on the number of households connected to the town water supply service, the total population of the town, and overall water supply coverage. This information is plotted on a graph to see the trend of the previous 10 years' variations and fluctuations to identify the necessity of an urban water demand management approach and propose near-future suggestions. Onsite assessments of the existing conditions of the water supply sources and public water points were conducted through critical observations and, if needed, immediate remediation suggestions for the town water supply utilities.

The paper is organized as follows: based on the overall background and objectives of the study, a series of assessments were conducted. First, an analysis was done on the accessibility and variability of drinking water sources and the current residential water demand management in the three towns. Next, the ongoing water supply services in the study area towns were assessed alongside a more general analysis of how Ethiopian water authorities are managing domestic water demand and the major determinants (responsible causes and factors). The service was then evaluated through a comparison with other global urban towns. Following these analyses, approaches for basic residential water management in urban towns of Ethiopia that prioritize sustainable and resilient development can be suggested.

### 3.3. Method of Data Analysis

The study combines qualitative research with empirical facts; such that to improve the future residential water using a suitable approach before finishing all the natural resources we have on the planet. Water supply and sewerage authorities' expert worker's interview responses are also used as important information in order to classify the differences in how the urban towns are managing their respective water utilization. And how and what similarities are there within the towns in managing urban drinking water demand. Using the information gathered from the interview, the current status of the water supply service situation and water demand management measures in the study areas was explained highlightly within this study by referring to other urban areas' water demand management measures, especially the developed countries. Such that what the gaps between these two areas demand management measures, and how to reduce the gap was one of the results expected from this study. Household-based (demand side) water demand management approach and general structural water demand management framework in developing countries were focused on.

Different freshwater availability determinants will be discussed and highlighted in this study. The past billed water consumption data and other quantitative determinants of water supply, such as demographic data, are needed to detect and compare the trend of water resources and the determinants. The number of customers, water supply coverage and population increase for the last 10 years' data trend were evaluated to see if water

demand management is needed for the future sustainable water supply service. Finally, the challenges of not implementing the water demand management measures were stated with the near future recommendations of easily implemented water demand management measures, specifically demand side based-management measures were shortlisted.

## 4. Results and Discussion

*4.1. Urban Water Supply Services and Availability Status*

Based on the analyzed interview data and observation, water supply services, and management activities in the three subject areas, the following assessments were made. Even though the information was assessed from three study areas, it is relevant and transferable to the other urban areas of the country. Because water supply management frameworks and regulatory concepts are the same throughout Ethiopia's urban regions, the majority of water supply services in Ethiopia's urban towns rely on groundwater through shallow wells, deep wells, and springs. Within our study areas, Arba Minch and Debre Birhan towns use groundwater sources, while Ziway town uses the Dembel River to source its water supply services. The general water supply services and the scheme of infrastructural statuses in each town's water supply were analyzed to identify what differences exist and what should be improved in order to enact sustainable water supply management and services. Here below is the detail of each town's water supply service information, which we get from feasibility study documents and interviews with the urban town's water supply utility experts.

### 4.1.1. Arba Minch

Arba Minch is divided into two sub-towns classified as an upper town called Secha and a lower town called Sikela. The water supply service in the town was constructed in 1963, but only to serve the upper town. In 1987 a modern water supply system was constructed to serve both the upper and lower towns. Currently, the water supply service of the town uses spring water, which discharges 95 Lit/s, and three boreholes, discharging about 75 Lit/s. The total volume of daily water supply from these sources is estimated at 9875 m$^3$/day. The source does not have the capacity to fulfill the demand of the town, which is nearly 11,500 m$^3$/day. Currently, the town is using around 10 surface pumps and 3 submersible pumps in its water supply utility. However, most of the pumps are not functioning due to their age and the challenges of maintenance, leading to water disruption. The utility has a total of 7 reservoirs with different capacities within the distribution system and a distribution pipeline length of 115 km and about 17 km transmission line. However, there are difficulties related to water quality deterioration, further exacerbated by the distribution and topography of the town.

The town utility service is working to reduce the non-revenue water by replacing more than 15 km of aged pipelines with new pipes. To increase the service and meet the demand of the customers, the authority is also constructing water points at different locations in the town. Thus far, 30 water points have been constructed for low-income areas, but this is not enough. The water points are open only for an average of 5 h per day, causing a further burden for women and girls who bear the responsibility for water fetching. Despite these obstacles, the town water utility is doing well in improving water quality. The Arba Minch Town authorities have established a laboratory for conducting the physical, chemical, and biological tests of water sources. They have also taken actions regarding consumer awareness of the activities started by the water demand management and NRW reduction.

### 4.1.2. Debre Birhan

Groundwater has been used as a source of water supply for Debre Birhan town since the construction of the water supply scheme in lately 1960s. There are 10 deep boreholes near the town (depths of 70–125 m) that distribute the water to the network system, which is divided into three zones or water basins based on the topographical nature of the town. Seven wells are found southeast of the town around an area called Dalecha, with a total

discharging capacity of 55 Lit/s, and four wells are found south of the town in the Baressa area, with a total discharging capacity of 48 Lit/s. The wells are equipped with electrically driven submersible pumps, which pump water for 19 h a day. Of the 10 total boreholes, seven of them have a standby generator for the time of electric power disruption. The wells produce a total volume of 77 Lit/s in 24 h, and the total water entered into the storage tank within 24 h is 6635 m$^3$. The distribution network has customer connection pipelines of approximately 92 km in length, with 5 km of bulk transmission pipeline. There are two ground-level reservoirs (2000 m$^3$ and 1000 m$^3$) and one elevated water tower of 250 m$^3$. There were a total of 30 water points around the town, working on average for 8 h a day, but three of the water points are not functional now.

The water loss within the distribution system because of leakage from the pipeline, illegal connections, and use of water before the meter are all major problems to the scheme. There are also some complaints from the residents around the north of the town that they are not receiving water due to weakened pressure within the pipe because of the topographical location of the area. For some members of the community, the Baressa river is another source of water used for bathing, washing clothes, cattle, and drinking water. However, due to the increase in economic activities of the town, especially hotels and industry, a significant pollution problem is damaging the river. Informal settlements and migration of people from the neighboring town due to the ongoing conflicts are causing extra burdens on water demand in the town. There is also no water quality laboratory established for the water supply utility of the town.

### 4.1.3. Ziway

Ziway (also known as Batu) is one of the towns located in the Great Rift Valley and found on the western shore of Lake Ziway. The town community gets its water supply from Lake Ziway, Dembel River, and three deep boreholes. A conventional treatment plant started serving the town in 2001, and a new treatment service called Tufa started operations in 2018. However, currently, the town water supply utility is getting its water supply source mostly from the boreholes. There are two ground reservoirs and one elevated reservoir within the distribution system of the town. It is composed of two 5000-m cube reservoirs and one elevated reservoir located around the prison center. The balancing reservoir has a capacity of 10,000 m$^3$ and is used to distribute the treated water to the consumers. In total, the distribution system is composed of a 76 km long pipeline and a 3 km long bulk transmission line. There are three deep boreholes fitted with hand pumps, and an NGO called Amref Health Africa rehabilitated two deep boreholes with 10 new public water points in 2019. Additionally, the NGO gave training for water management committees and water caretakers in the area. Lake Ziway supports a multitude of uses, including irrigation, fishing, water supply, and recreation. The biggest commercial floriculture investment in the country is located on the shore of Lake Ziway and depends primarily on its water. Human impact on the lake water quality and quantity and over-extraction of water is putting the lake under stress, and the government is not giving enough attention and effort to watershed protection. There is a lot of deliberation about floriculture industries located at the shore discharging untreated waste directly into the lake; however, excessive fertilizer and pesticide residue continue to deteriorate water quality as well as the aquatic life [25].

The urban area was degraded around the end of the 1990s and had no vegetation cover. Deforestation and flooding were major problems in these areas. Now the area is covered with grasses and vegetation, and people have been made aware of catchment protection. Although soil erosion has now decreased, there is still a shortage of drinking water in the area. Excessive fluoride concentration >1.5 mg/L consumption in drinking water, which includes both groundwater and Lake Ziway springs, caused a health problem in the communities. The government and the municipal water utility give little attention to water quality. The city's water production capacity over the past decade shows no significant changes, and the water distribution system infrastructure is not being modernized; thus, it can only serve the same customers that it served 10 years ago.

*4.2. Summary of Findings from Interview, Document Review and Its Elaboration*

Generally, the evaluation of the three water supply services shows that there is a shortage of supply that cannot meet the current demand of the population. Additionally, some of the residents in Ziway and Debre Birhan are using water from unprotected lakes and rivers without any treatment. Even though Ethiopia is known as the "water tower of East Africa" with a big amount of surface and subsurface water resources, public water is still supplied from unprotected lakes, rivers, and wells that are unsafe and polluted in both urban and rural areas [26]. Some of the complaints from the town residents report that the distribution of water supply is not fair, as some areas are getting more water while others struggle. Arba Minch and Debre Birhan both have issues with water pressure within the distribution system because of the topography of the towns. Research shows that water resources are distributed unevenly within Ethiopia because of the topographical and geographical landscape, with highland areas having plenty of rainfall and lowlands faced with desert climate conditions. Ziway town is one of the most affected areas because of the absence of rainfall, land degradation, and flooding [27]. The variability in topography creates challenges for accessing water resources easily and delivering water to all consumers [28].

Within the three study towns, we determined that most of the people and consumers have limited awareness about where the water comes from and through what processes the water passes before it reaches their tap. They only know that the source of the water is spring water or groundwater, such that they call the tap water "God's water" because it is available anytime and is considered free of charge. People believe water that is bottled and sold in a market is the only water that should be charged for, whereas all other water should be easily accessible at any time. Another problem with the water authority in all three research regions is their reluctance to maintain pipe breaks inside the distribution system, often taking longer than a week to fix a pipe break. During this time, the nearby neighborhood would not have access to water. The acquisition of a new pipe was cited by utility officials as the cause of the delayed maintenance of pipe breaks. It is important to make people aware of the relationship between environmental changes and the sources of water for drinking, including what might happen in the coming few decades if we continue overusing and mismanaging these resources.

*4.3. Reviewed Determinants of Water Demand*

Identifying the specific determinants of residential water demand in order to address future management options and strategies is a crucial concern. The most common factors affecting residential water demand are the size of the town or city, living standards, climatic conditions, industrial and commercial activities, quality of water, a system of water supply, a metering system, and a system of sanitation in the town. The increase in population, the number of people connected to the town water supply service, urbanization, and water supply coverage are the determined potential determinant raised to be discussed within this study. These determinants are selected based on the country's growth standard and the status of water supply and sanitation facilities.

4.3.1. Population

The population of all three study areas has grown over the past 10 years due to the migration of people from rural to urban areas. The town of Ziway is relatively small, while Arba Minch and Debre Birhan have relatively higher populations. The data shows that Ziway and Debre Birhan areas have had a population increase of more than 50% in the last 10 consecutive years. As the population grows, so does the demand for water supply, which represents an additional burden on municipal waterworks; as the urban population grows, so will the number of users. Figure 2 below shows the population trend over the last 10 years in the towns of Arba Minch, Debre Birhan and Ziway. The percentage increase in population is 45%, 57%, and 50% for Arba Minch, Debre Birhan, and Ziway, respectively.

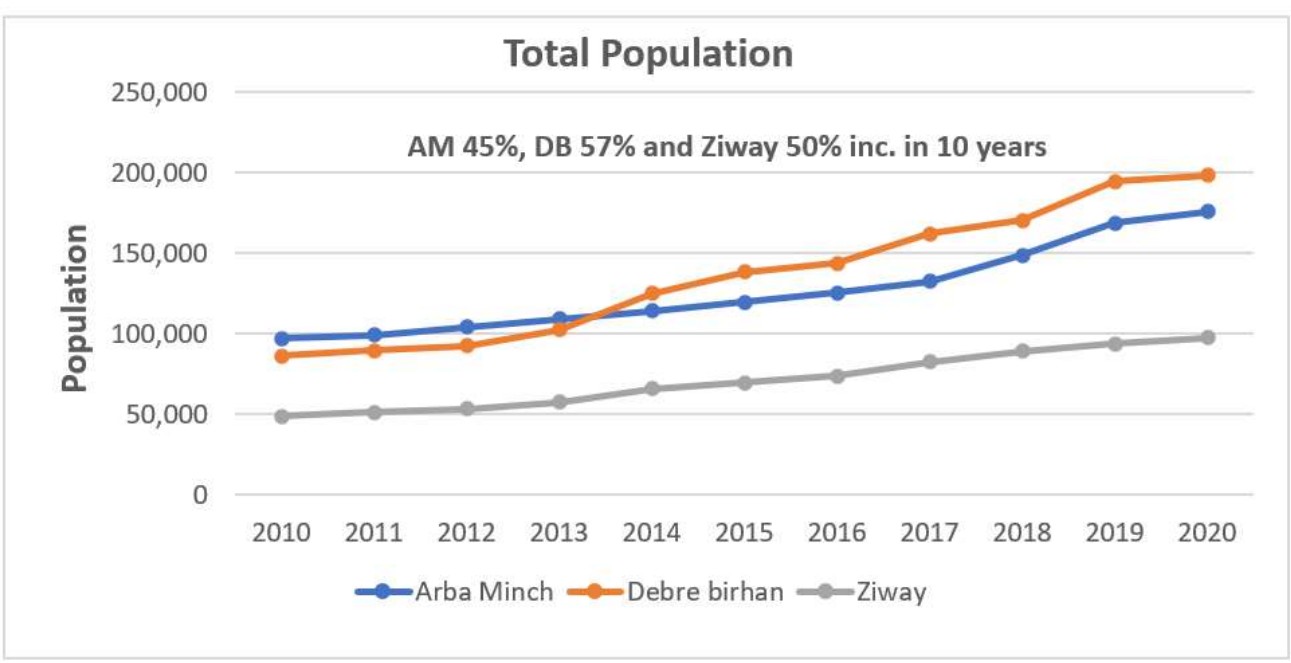

**Figure 2.** Population increase trend of the three towns for the last 10 years.

4.3.2. Customers

Increasing population density has put high pressure on existing water sources in Arba Minch, Debre Birhan, and Ziway. In the past 10 years, the average number of customers connected to the town water supply service was 48% for Arba Minch and Debre Birhan and 44% for Ziway. The residents of the towns can access water at least twice a week. They fetch water from a tap and store it in their house when there is no water supply. The quality of water can be adversely affected by this method of storing it. Data shows that the current water production amount from the town water utility is less than what the population demands: households are only getting 2 to 3 h of water supply in a day with little reliability in each urban area studied. The lack of infrastructure and current management practices are more to blame for the towns' problems than the lack of water. That the distribution system is quite old and needs to be replaced by a bigger pipe to compensate for the demand of consumers is one of the problems raised by the Arba Minch and Debre Birhan water authorities. The infrastructure of the water supply system was not updated as the number of customers increased. Water demand management can be used to improve the current system. The number of users connected to the town water supply service is shown in Figure 3.

4.3.3. Urbanization

The term urbanization used in this research refers to the expansion of the city geographically. One of the basic developmental infrastructure components for urban areas is water supply service, which should increase at the same time as the expansion of the town or urbanization. The opposite is happening now in three study areas; urban water supply infrastructure is not updating itself even as the town is expanding. All of the water supply systems functioning now were constructed more than two decades ago and have since had very limited or no update in structural or service capacity. Urban-rural migration and informal settlements are some of the factors increasing urban population growth in Debre Birhan and Arba Minch towns. While Ziway is not expanding as compared to the other urban areas, it is instead enhanced by an industrial zone comprised of different flower farming industries. The three study areas are technically urban when compared with the surrounding rural areas, but they all lack the basic infrastructure needed for an urban area. Recent urbanization-related studies in Arba Minch and Debre Birhan revealed that



high growth rates of the urban population will continue, resulting in a higher level of urbanization within a few years. The close proximity of Debre Birhan to the capital city and the conflicts happening in the northern part of the country have also increased the number of people moving to the town.

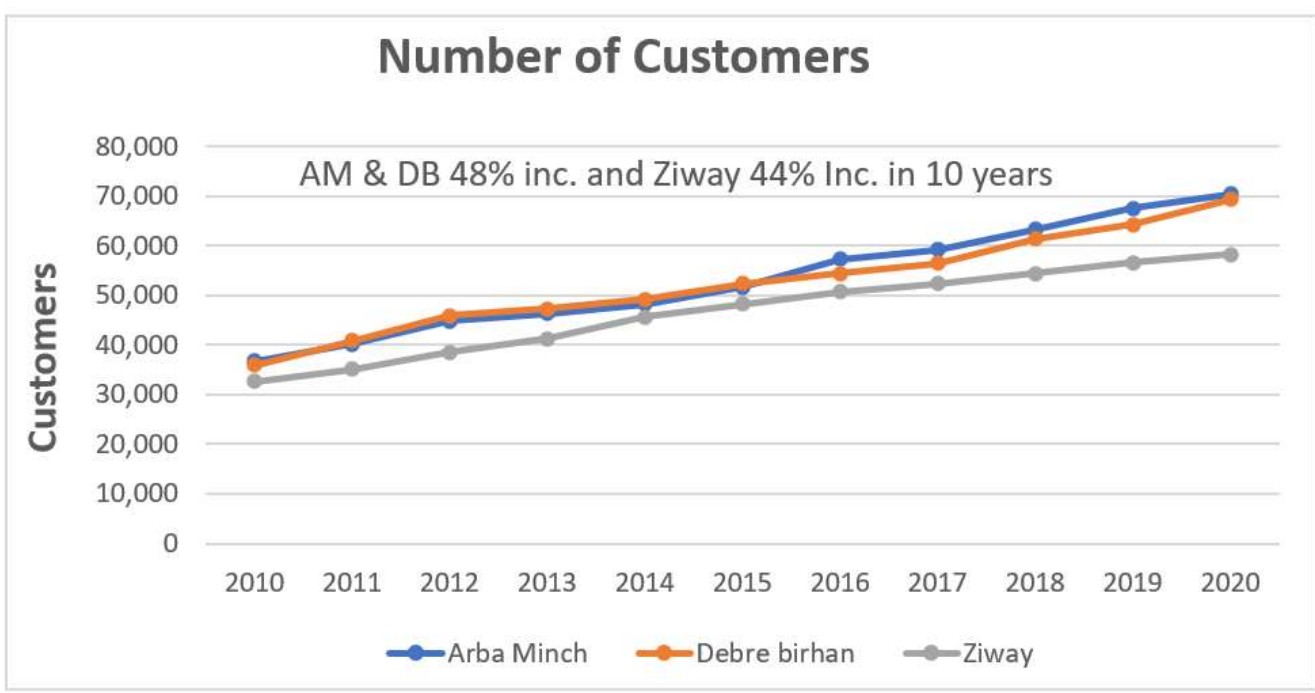

**Figure 3.** Trends in the number of consumers for the last 10 years.

A research study done by [29] shows that Ethiopia is under a water stress condition, with current water availability per person per year measuring at just 1109 m$^3$. Arba Minch, Debre Birhan, and Ziway all exemplify water-related challenges. A previous investigation into the water management process has largely excluded assessments of the impacts of varying forms of urban development patterns on water demand. Hence, water-centric management strategies, such as water-smart community development, water-sensitive physical planning, water-smart technology, and a water-sensitive legal framework, are required to address the ongoing water risks [29].

### 4.3.4. Water Supply Coverage

The percentage coverage of water supply is one of the best indicators to describe the standard of water supply service. However, there has been no significant improvement in the expansion of water supply coverage within the last 10 years (the water supply coverage of the three urban areas for the last 10 years can be seen in Figure 4). This illustrates around 40% coverage for Arba Minch and Debre Birhan and an average of 65% coverage within 10 years for Ziway town. This figure depends on population increase as well as the number of customers registered at the town's water supply authority and using the town's water supply connection. Because the number of customers is increasing every year alongside the rise in population, the percentage coverage does not show that much change within a year. This result reveals that Arba Minch and Debre Birhan are vastly growing in population, and the demand for water is consequently increasing, while Ziway is not expanding or increasing its needs at the same rates as the other two urban areas. Despite the differences in growth, in all three study areas, consumers receive water on average 3 days per week, and only for a specified time, mostly in the morning and evening. Customers must fetch and store water for the times that there is no access and boil this water for drinking.

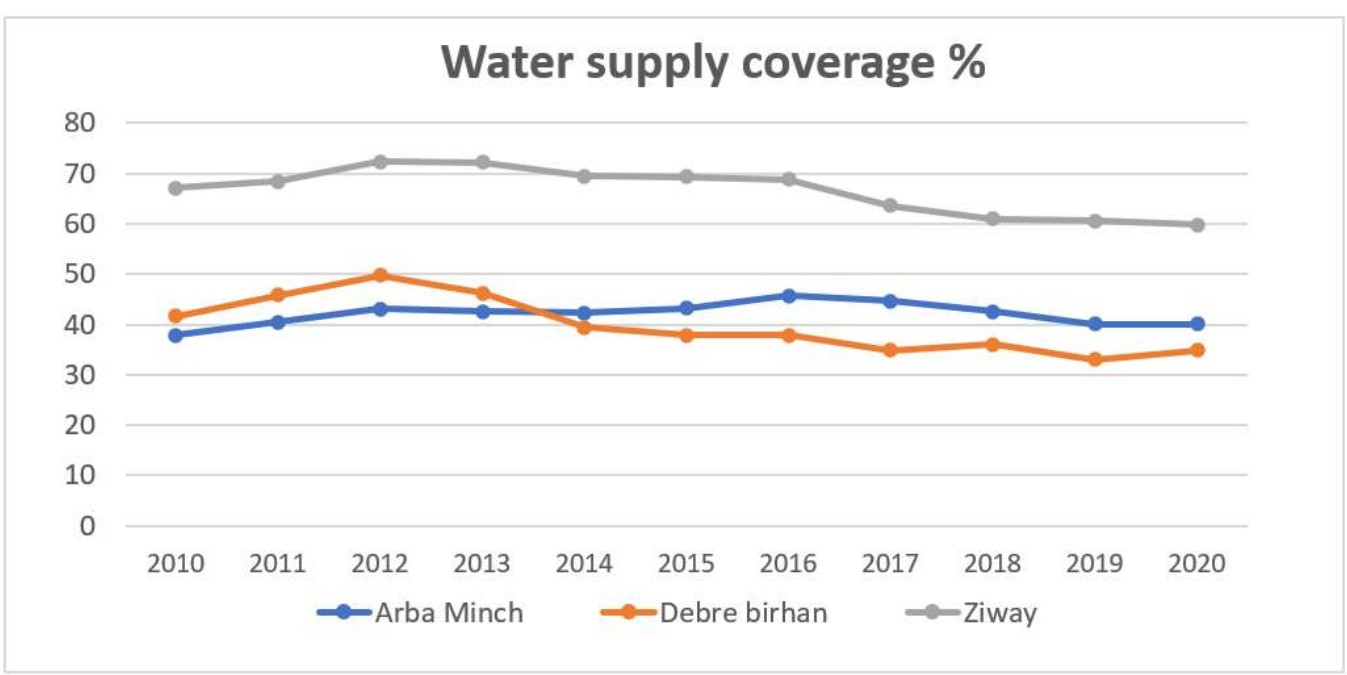

**Figure 4.** Water supply percentage coverage of the past 10 years at the three urban towns studied. Source; water supply and sewerage authority of the urban towns.

General water supply coverage in Ethiopia has increased significantly over the last two decades. According to government data, water supply coverage has risen from 19% in 1990 (11% rural, 70% urban) to around 69% in 2010 (66% rural, 92% urban) [30]. The expansion of the urban areas and the limited coverage within the newly expanded areas causes all residents of the town to suffer from shortages of water. Coverage estimates from the Joint Monitoring Programme (JMP) are significantly lower, with access to drinking water reported at 44% in 2010 (34% rural and 97% urban) [31]. Despite the differences between the JMP results and the government data, on the ground, there is still a huge gap between water demand and supply. In addition to the secondary urban towns in Ethiopia (our study areas), the residents in Addis Ababa also suffer from an acute shortage of water and service disruptions and thus get water supply just once a week and sometimes once in two weeks [32].

Water supply in most of the urban areas in Ethiopia is equally lacking water utility recovery and is not progressing towards its goal. Customers are not served at the standards required, and their expectations are not met due to problems in the sector's governance, drinking water coverage, and the discontinuity of service because of limited maintenance.

As the above discussion details, the size of the population and number of customers connected to the town water utility is increasing, urbanization is growing, and water consumption is increasing, while water supply coverage is decreasing or not growing, and freshwater resources are also diminishing. However, there are no other actions taken from the government side or from the consumer side to maintain the sustainability of the resources. As indicated in Figure 2, the number of customers is increasing while the supply coverage is decreasing (as shown in Figure 3). The increase in population and urbanization is one of the factors causing shortages of water supply services. The number of displaced populations should be considered as a factor in increasing coverage as people move towards urban areas from all around the country because of conflicts and violations. Rising populations put extra pressure on water supply services. Urban water authorities must work hard to bring sustainable water supply services by focusing on solving their service inadequacies, such as prioritizing repair for old and leaking pipes.

The relationship between water extraction and water availability has turned into a major stress factor in the urban exploitation of water resources. This is expected to be

continued unless we provide some important mitigation measures on water demand to address both climate and non-climatic changes. Tapping the water resources for drinking purposes is a challenge due to financial constraints and technical problems. Despite these challenges, Ethiopia has plenty of water resource potential. The important thing for the future of the country is to identify the most effective strategies to unleash this potential and meet national development objectives. There must be full knowledge of water resource management issues and environmental protection merits at all levels of governance, including the local communities and residents of the town. The future of any country depends on the availability of freshwater resources, and water is a strategic commodity and resource that can intensely affect sustainable development. Urgent demand-side water management actions are recommended to address the availability and sustainability of water supply services. Strengthening ongoing demand management evaluation in order to fill the gap in urban areas is the next move for maintaining future sustainable water supply services.

*4.4. Water Demand Management Practices and Progress*

The water demand management practices observed in Arba Minch, Debre Birhan, and Ziway reflect the water demand management practices all over the country's towns. Ethiopian water resource management and development strategies depend on economic and non-economic measures. According to some reviewed documents, the national water management development (2000) strategy initiated by the government of Ethiopia gives more attention to the use of non-economic mechanisms for water demand management than economic measures [33]. These reports suggest that using water pricing to reduce water demand is not acceptable as the national water demand management option but should be replaced by work on community-based demand-side management measures. The assessment was done in the three subject towns, and relevant literature shows that the scarcity of residential water supply needs an urgent response that will promote demand-side water management measures.

One of the demand management practices increasingly used in Arba Minch, Debre Birhan, and Ziway is to create extra water sources, such as private wells. This kind of practice is common in all urban areas of Ethiopia, but its success depends on the topography of their location and the presence of groundwater availability. Accordingly, even within towns, there are zones where hand-dug wells are not an option because they cannot reach a groundwater table with such shallow digging. Although the community has little awareness about water demand management, they are forced to manage their own resources because the water supply they are getting from the government is not enough to meet their demand. The residents don't have secure access to the town's water supply service, so they search for their own solutions. As a result, some households have both private wells and tap water connected to the town water supply authority. Those households that don't have a private well in their backyard are forced to fetch water whenever the tap is running and store it in a container for the times when there is no water available. Some households don't have either type of access and are forced to fetch water from nearby surface water sources and use local treatment methods like boiling or filtering with clean wool.

Even though there are water resources management strategies theoretically known by every governmental water utility, on the ground, there were no activities done by local governments or water utilities to implement demand management approaches. According to the WWAP report [5], water management is not only a technical term but also encompasses various scientific disciplines and activities. It requires measures and practices which might change people's perception of using water. Thus far, there have been no activities done by the government or urban water utility officials' side to teach the community about water demand management. Teaching and awareness-building within the community are necessary to change behavior and adequately manage and use scarce water resources.

An Integrated Urban Water Management (IUWM) approach is one of the management tools that emerged recently in Ethiopia to help urban towns achieve sustainable economic,

social, and environmental goals. IUWM promises a better approach than the current system, in which water supply, sanitation, stormwater, and wastewater are managed by isolated entities, and all four are separated from land use planning and economic development. However, the implementation of the approach doesn't bring better solutions for demand management in urban water supply services. These approaches require infrastructure such as a proper continuous water supply distribution system and wastewater and stormwater discharging management systems which are not available in Ethiopian urban areas. Currently, there is no proper wastewater management system; all water disposed of in bathrooms, kitchens, and residential buildings is stored in private septic tanks near the houses. Given these conditions, it is difficult to consider water demand management approaches as there is not enough water for full access, nor are there sufficient facilities for wastewater discharging. Further exacerbating the problem, there are multistorey buildings being constructed in all three study areas, which need sustainable water supply services. In the past 10 years, a number of common residential multistorey buildings have been built to compensate for the shortage of residential houses in the town. Governments are building hundreds of residential buildings to eradicate the housing shortage within the town, but the supply of residential water available now is not enough and will not meet the increased demand.

### 4.5. Other Countries Urban Water Demand Management Experiences

Water demand management is one of the most serious challenges facing major cities all over the world. Fresh water sources for supply services are rare, and other water sources are polluted and must be treated at a high cost [34]. In addition, the volume of wastewater from sources such as households, commercial and institutional operations, and industry is growing due to urbanization and population increase. All countries have their own way of dealing with freshwater demand management as well as wastewater management systems either at the governmental level or community level.

In developing countries, which must manage large informal settlements, water utilities that have effectively reduced demand often have community-based water-conservation programs [35]. Community-based approaches for water demand management used in slum areas of developing countries in Asia, such as using single bulk meters and using the help of local NGOs, have arguably had significant impacts on demand, particularly in reducing NRW. In Indonesia, accountability was given to the local governments to manage water resources and sustain the resources they have, with positive results [36]. Malaysia, the Philippines, and Indonesia have all made important institutional reforms to promote accountability [35]. Collectively, the experience in South-East Asia strongly suggests that leadership plays an important role in improving the performance of urban water utilities, particularly with respect to demand management.

Rainwater harvesting and leakage management control programs are being promoted in many cities in India, including New Delhi and Chennai, as a measure to reduce the water demand from the existing piped water system. Furthermore, in an attempt to reduce water demand, many water utilities and environmental agencies in developing countries are conducting awareness-building campaigns and school education programs on water conservation [37]. The WDM strategy used in Windhoek, Namibia, consisted of policy issues (including water pricing), information campaigns, legislation, and technical measures. As a result, the residential water demand level was lowered from 201 to 130 L/capita/day between 1992 and 1999 [38].

Most developed cities have been actively pushing demand-side management activities. Toronto, for example, has invested in programs such as ultra-flow flush toilet incentives and industrial water capacity redemption with the goal of reducing peak water demand by 15% [39]. Other cities in North America have also used different successful water conservation. Some of the conservation measures they used include school water conservation posters and limerick contests, an expanded retrofit program, toilet replacement incentive

projects and rebate programs, the implementation of water checkups for large residential water users, and enhanced in-school curriculum-based education [40].

Environmental protection from wastewater disposal to fresh water sources and water resource management are highly interrelated. Environmental protection regulations indirectly discourage the pollution of clean water sources and make the community aware of their natural resources and water supply sources. As an example, Berlin enacts water demand control and environmental protection, keeping their available water resources clean and reusable through awareness creation on pollution and reducing water use. General awareness creation on environmental protection can support the management of available water resources to be sustainable [41]. Even though the Ethiopian urban community has limited awareness about wastewater disposal management, the government should teach local users how to protect their environment. There are no wastewater management systems in many towns, as well as no wastewater management policy and no regulations on wastewater disposal. Therefore, a proper strategic policy is needed to improve water service delivery and wastewater management policies. Despite the differences in the socioeconomic level of people living in cities of developed and developing countries, water demand management measures need to be formulated to consider the local situations of the community.

*4.6. Future Potential Solutions of Water Demand Management in Urban Areas of Ethiopia*

Recent recommendations have generally centered around practical ways of reducing the supply-demand gap in relation to water and more effective policies and policy instruments. These have included water saving and conservation, pricing mechanisms, and water investments. Many of the recommendations were specific to actions at the level of the city water authority, which had responsibilities to supply and distribute water. Traditional approaches to water resource management are not considered unsustainable. Demand-side water demand management approaches have also not been given fair attention so far. Water resources should be managed in an integrated manner. Although water demand management tools such as pricing and non-pricing bring positive changes in water resource management, in sub-Saharan African countries like Ethiopia, urban areas have not yet shown a significant change in water demand management. There should be modifications based on the standard of the country in order to manage water demand by including a local demand management approach.

Across the interviews with water utility officials of the three study areas, a weakness in policy implementation and general inefficient policy across levels of government sectors was observed. Policy instruments that would encourage the use of water-efficient technologies and policies supporting full-cost recovery in water service delivery, which is important for water demand management, were lacking in all the town utilities. Such policies would be immensely helpful in addressing the needs of the rising number of condominiums and cooperative houses in urban towns. Also, there is no monitoring system for controlling the type of water fixtures installed within the building and how much water they require. Another complaint raised by the Ziway and Debre Birhan water supply utilities described a long procurement process for purchasing any materials needed for maintaining the distribution system. Both controlling non-revenue water and replacing old pipes in the distribution system require urgent solutions to ensure water is not wasted. Therefore, the procurement sector in water utilities should be given priority attention.

Ethiopia had no separate budget allocation for the water supply sector until 2002, when independent water supply and sewerage offices were established. Prior to this, the water supply sector received its budget share from the local governing office, often much less than the shares for agriculture, education and health. Even though the budget for water supply has increased in recent years, it is still insufficient and skewed towards recurrent expenditures at the cost of capital investment. The allocated budget is only used for the central urban towns, and the local surrounding community doesn't benefit. Some officials are reluctant to alter this share, as they presume the surrounding areas are

getting water supply services from NGOs. Another important issue observed from the assessment was that the organization structure for water resource management is very centralized, while sectoral coordination, stakeholder participation, and decision-making remain unsatisfactory. The number of water supply professionals and skilled manpower available in all three urban towns studied was insufficient in numbers. Generally, adequate attention was not given to the water supply sector.

Environmental protection and awareness creation could bring a real change in water resource management. As a result of rapid population and industrial growth, the demand for water by the domestic, commercial, and industrial sectors continues to rise while water availability continues to decline, further complicated by competing users, environmental degradation, and climate change [9]. A recent study by Timotewos MT et al. [42] demonstrated that the impacts of topographical distribution and other anthropogenic activities on water demand are more significant than climate change in Arba Minch, Debre Birhan, and Ziway. The assessed water demand management measures from other nations, as shown in Table 1, mostly focus on educating the community to use less water. Therefore, teaching people to save water is the primary water management mechanism we need to focus on to reduce anthropogenic effects on water demand. The federal government and local town water utilities should work on awareness creation among the community to save water. Water saving should be connected to the culture of the people and advertised through engagements with stakeholders or community gatherings and school curriculums. One of the solutions for declining water supply services is maintaining sustainable sources of water supply by using sources of water from deep aquifers. Arba Minch and Debre Birhan are doing an exemplary job of increasing their supply for the community and growing their local water sources by conveying fresh water from deep aquifers.

**Table 1.** Reviewed studies on strategies for water demand management.

| Country | Strategy and Approach Used | Result Obtained | Source |
|---|---|---|---|
| Asia (Indonesia, Malesia, Philippines) | Single bulk meters, active NGO engagement in water service expansion, leadership reforms to promote accountability | Reduced NRW | [35] |
| Indonesia | Giving full accountability and authority to local government | A positive result in water demand management | [36] |
| India (New Delhi and Chennai) | Rainwater harvesting, leakage management control program, awareness building campaign, school education program | Reduced NRW and reduced water demand | [37] |
| Windhoek, Namibia | Water pricing, information campaigns among the community, legislation | Water demand lowered from 210 to 130 Lit/cap/day between 1992 and 1999 | [38] |
| Canada, Toronto | Ultra-flow flush toilet incentives and industrial water capacity redemption | Reduced peak water demand by 15% | [39] |
| USA | School water conservation posters, limerick contests, expanded retrofit program, toilet replacement incentive project, school curriculum-based education | Reduced water demand | [40] |

Finally, one of the major goals of this research is to propose a residential water management approach from the demand side of management. There is a multitude of limitations on efficient management beyond the presumption of water resource scarcity, including a lack of coordination, absence of adequate authorized institutions to manage the supply, rapid population growth, and technical, financial, and political problems. Thus, the main purpose is to propose and show a way forward for the local water utilities to adopt a water management measure that fits the community's living system. Demand-side management has long been neglected, and water demand management is overly focused on supply-side

management approaches. Growing research in sustainable water resources management has demonstrated that demand-side management is a better approach. Education and awareness building on water efficiency and water conservation methods at the consumer's side should be promoted among all stakeholders of urban areas. WDM should be promoted both as a community culture and as an option for the water supply service provision in urban areas.

## 5. Conclusions and Policy Implications

This research study has explored the water demand management measures of developed countries alongside water demand measures applied in three urban towns in Ethiopia. Water pricing measures are a primary tool in developed countries and play an important role in addressing water demand and saving water resources. However, these approaches would not have the same impact in developing countries like Ethiopia if directly applied.

As identified from this study, Ethiopia is under the influence of urbanization and urban population increase. Water development strategies must unquestionably follow urban growth projections. Water demand management approaches are fully focused on supply-side measures, which are incompatible with-long term development. Consequently, existing water resources will be depleted if important water-saving measures are not adopted in the near future. We must change our approach to consider the demand-side options, not only supply-side management measures. This can be accomplished by lowering the water loss of leaking pipes in the distribution system, establishing awareness by educating the community about water demand management, and introducing water-saving facilities for the newly constructed condominium houses. Unequal accessibility of water supply services is a major problem households are facing within the three studied areas. There is an issue of inequitable distribution of water among the residents of the town and also new areas which cannot access any water from the town utility. In addition, a large variation in water consumption was noticed between households having proper water supply facilities within their premises and those without access. One of the approaches governments are using now to cope with the scarcity or shortages of urban water supply services is promoting private hand-dug wells. Relatedly, Kou L et al. [43] stated in their study that the prevention of future water shortages requires the implementation of water-saving measures as well as the use of new water supplies. This could be one option to reduce the shortage of water supply services and increase the household's water consumption. Inversely, the management and control of water quality would become more complex if sources were to be decentralized, which could lead to increased community health problems. Therefore, the government should be actively involved in educating the community on onsite water treatment mechanisms as well as tools to manage the water sources they have.

There is a gap from the government side also in managing specifically residential water supply and developing policies related to water use. As urbanization increases, the issue of new multistorey residential buildings and condominiums is also increasing. As a result, this study encourages developmental policy geared at the residential buildings and apartments that demand technologically and economically feasible water facilities like water-saving flush toilets, faucets, and showers. Additionally, the greywater disposal from these residential areas should be managed so that it can be reused for different purposes like gardening and toilet flushing since the greywater disposed of buildings can represent as much as 70% of the total household wastewater volume [44]. Having a policy on water use in communal residential buildings and developing greywater reuse technologies would allow governments to save about 20–30 percent of water used in the buildings.

The findings from this study helped us to identify measures that could help town water utilities to manage residential water supply services and better support users. However, there is a lot of work needed to be done by the residents and government side. Strategic planning is a key aspect of a successful water demand management strategy. There is a lack of awareness of water resource management, specifically water demand management, from the user's side. There must be action from the government side on policy development

toward water use and environmental protection. Structural management arrangements should be given attention, and water utility officials with professional knowledge of water resources and an educated, skilled force should be hired. The procurement system of water utility materials should be revised so that operation and maintenance can be done to maintain sustainable water supply services and reduce the non-functionality of water schemes. Recommendations for future research should mainly focus on water demand management policy and environmental protection strategies in order to attain a sustainable water service. Aged water supply infrastructures, high population growth rates and corresponding water demand, high water losses in the distribution system, poor water management, and a lack of proper institutional framework characterize urban water supply systems in Ethiopian towns.

**Author Contributions:** M.T.T. and B.M.B. developed the research idea and methodology. M.T.T. collected the required data and performed the analysis and manuscript writing. M.B. followed up on all procedures, supervised the study, and edited and reviewed the manuscript. All authors have read and agreed to the published version of the manuscript.

**Funding:** This research is part of the DAAD-EECBP Home Grown Ph.D. Scholarship Program under (EECBP Homegrown Ph.D. Program-2019). And also funded by the Open Access TU Publication fund of the Technical University of Berlin, Berlin.

**Data Availability Statement:** The original data used in this study are available from the authors on reasonable request.

**Acknowledgments:** We would like to thank Arba Minch, Debre Birhan, and Ziway towns' water supply offices for providing valuable datasets for this study. The authors would like to thank the three anonymous reviewers for their valuable comments, which were helpful in improving the paper.

**Conflicts of Interest:** The authors declare no conflict of interest.

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
