# Peer review of "Setting a Baseline Residential Water Demand Management Solution in Urban Towns of Ethiopia"

_water, doi:10.3390/w15050930_

Round 1

Reviewer 1 Report (New Reviewer)

Setting a baseline residential water demand management solution in urban towns of Ethiopia

I have read your manuscript with great interest. While I think the subject of the manuscript fits well with the scope of the journal water, the writing style and the scientific arguments are not at the level expected for a publication within this journal. The authors try to address a hot topic that is important for the community. The manuscript is very poorly written. There are entire sections where it is nearly impossible to follow the information outlined. The manuscript needs very fundamental editing and improvement in scientific writing before anyone can look at the science of the report a little bit deeper. This MS has the fair potential to publish in the water after revision. It focuses on Setting a baseline residential water demand management solution in urban towns of Ethiopia. The figures need some tidying up and captions expanded enough that the reader can understand and be guided on what to see on them without actually reading the main text.

Abstract

 document analysis, what it means which document you have been analysis

The assessments at the three study regions demonstrate that while the percentage of 20 water supply coverage hasn't changed significantly over the previous ten years, both the number of 21 customers and the overall population growth have increased by nearly 50%

 This is mean the finding of your manuscript  but you did not tell about the data set and methodology on what basis you conclude this

Introduction

The section needs major improvement. the Introduction is rather a chaotic section where much fragmental information is presented, in many cases not logically linked,

Your article must answer the following basic questions:

*       The problem statement under investigation, methodology, and results/and discussion must be aligned.

*       What are your results?

*       What are the implications of the results?

*       What do you recommend as a further study for others?

I would like to ask you to cite a similar latest publication and improve your introduction. It is suggested to add other underdeveloping countries examples to make this paper readable internationally

Saleem, Arfa, Imran Mahmood, Hessam Sarjoughian, Hasan Arshad Nasir, and Asad Waqar Malik. "A Water Evaluation and Planning-based framework for the long-term prediction of urban water demand and supply." Simulation 97, no. 5 (2021): 323-345.

Ahmad, Naveed, Sikandar Khan, Muhsan Ehsan, Fayaz Ur Rehman, and Abdullatif Al-Shuhail. "Estimating the total volume of running water bodies using geographic information system (GIS): a case study of Peshawar Basin (Pakistan)." Sustainability 14, no. 7 (2022): 3754.

Sadr, Seyed MK, Line T. That, Will Ingram, and Fayyaz A. Memon. "Simulating the impact of water demand management options on water consumption and wastewater generation profiles." Urban Water Journal 18, no. 5 (2021): 320-333.

Sohail, Muhammad Tayyab, Asrar Hussan, Muhsan Ehsan, Nadhir Al-Ansari, Malik Muhammad Akhter, Zaira Manzoor, and Ahmed Elbeltagi. "Groundwater budgeting of Nari and Gaj formations and groundwater mapping of Karachi, Pakistan." Applied Water Science 12, no. 12 (2022): 267.

Figure 1. Study area location map

It is suggested to let you know about reading the data source of this figure.

3.2. Methodology

This section looks like a lab report. It is suggested to properly write this section in a proper scientific way and provide a proper reference.

3.3. Method of Data Analysis

This section looks like a lab report. It is suggested to properly write this section in a proper scientific way and provide a proper reference.

4. Results and Discussion

This section needs significant improvement. It is suggested to remove unnecessary information and rebuild this section in a proper scientific way. The current form is not acceptable.

4.1. Urban water supply services and availability status

What is the source of information for this section?

4.1.1. Arba Minch

What is the source of information for this section? You present numerical values but not tells the reader about the source

5. Conclusions and Policy Implications

Why? How was this conclusion reached?

Author Response

Dear Reviewer,

Thank you for your most valuable comments all the comments and suggestions given by you are very important to enhance the quality of the manuscript. we have revised all comments carefully and included them in the revised manuscript.

Reviewer 2 Report (New Reviewer)

Please find my comments attached.

Author Response

Dear Reviewers,

Thank you for your valuable comments and suggestions. All the given comments are helpful to enhance the quality of the manuscript. we have carefully revised and included them in a new version of the manuscript.

Round 2

Reviewer 1 Report (New Reviewer)

A agree with revision.

This manuscript is a resubmission of an earlier submission. The following is a list of the peer review reports and author responses from that submission.

Round 1

Author Response

Dear Reviewer,

Thank you for your most valuable comments to enhance the quality of the manuscript. As per your comments, we have been revised all comments in the revised manuscript carefully. the details of the responses is attached below.

kind regards

Reviewer 2 Report

Dear Authors, I liked your article a lot and have just some minor remarks: 

Line 190: Can you please explain why your findings can be transfered to other Ethiopian towns?

Figure 4: Please insert the reference (Coverage in % - of population? Households?)

Line 478, 602 and 654: Please insert Author's name before the citations

Kind regards!

Author Response

(The authors gave the same response as above.)

Round 2

Reviewer 1 Report

The authors have made all the necessary changes in the revised version of the manuscript.